# Beyond Per-Question Privacy: Multi-Query Differential Privacy for RAG Systems

## Abstract

Retrieval-augmented generation (RAG) enhances large language models (LLMs) by retrieving documents from an external dataset at inference time. When the external dataset contains sensitive and private information, prior work shows that without any protection, the RAG system has the risk to leak this information, which might hurt the data owner's privacy. The existing work has studied protect the information not leaked from a single-query from RAG under differential privacy (DP). In this paper, we focus on the more practical setting where the DP is extended to multiple queries from RAG. We propose two new algorithms that ensure DP in multi-query RAG. Our first method, MURAG, applies an individual privacy accounting framework, allowing the privacy cost to depend on how often each document is retrieved rather than the total number of queries given. Our second method, MURAG-ADA, further improves efficiency in the individual privacy accounting framework by adaptively releasing private query-specific thresholds for more precisely relevant document selection. Experiments across four question datasets and three LLMs show that our methods answer 100 queries under $\varepsilon = 10$, while baseline methods require $\varepsilon = 1000$ for comparable utility. We also highlight scenarios where our approach outperforms fixed-threshold baselines and discuss when individual accounting is preferable to subsampling-based techniques.

## 1 Introduction

Retrieval-augmented generation (RAG) has become a popular approach for deploying large language models (LLMs) in real-world applications. A core feature of RAG is its reliance on an external dataset as the primary knowledge source at inference time. For example, a medical RAG system may retrieve historical patient records to answer clinical questions more accurately. However, such external datasets often contain sensitive or confidential information. In domains like healthcare or law, the retrieved content may expose private records, raising serious privacy concerns. Prior work has shown that RAG systems without proper safeguards are vulnerable to information leakage [Naseh et al., 2025, Liu et al., 2025, Anderson et al., 2024, Li et al., 2025, Zhang et al., 2025, Zeng et al., 2024a, Jiang et al., 2024, Peng et al., 2024], compromising data owner privacy and user trust.

Differential privacy (DP) is a widely adopted framework for providing rigorous guarantees on individual data protection. Recent work [Koga et al., 2024] has proposed DPSparseVoteRAG, a RAG system that ensures the generated answer satisfies DP with respect to the external dataset, for *a single user query*. Empirical results demonstrate that this approach outperforms a strong baseline using a public LLM without RAG, while achieving an $\varepsilon$-pure DP guarantee with $\varepsilon \approx 10$.

In realistic deployments, users may issue a sequence of queries, allowing an adversary to aggregate responses and potentially infer sensitive information about the external dataset. A naïve approach that applies DPSparseVoteRAG to each query and relies on standard differential privacy composition

quickly exhausts the privacy budget. As shown by our experimental results (Figure 2), to achieve reasonable utility this approach achieving may require a privacy budget as large as $\varepsilon = 1000$, which is generally considered meaningless. This raises a key question:

*Can we design a differentially private RAG algorithm that handles hundreds of online queries while ensuring both strong utility and meaningful privacy?*

We answer this question affirmatively and summarize our contributions below.

- **Novel DP Multi-RAG Framework.** We present a novel framework for multi-query differentially private RAG. Our contributions include: (1) bypassing composition across queries using individual Rényi filters [Feldman and Zrnic, 2021], which significantly reduces the ex-ante privacy budget. To the best of our knowledge, this is the first use of privacy filters in the RAG literature; (2) enhancing both privacy and utility through threshold-based screening of relevant documents. Notably, the framework also adopts a modular design, allowing integration of any private single-query RAG algorithm as a subroutine. Additionally, the framework adopts a modular design, allowing the integration of any private single-query RAG algorithm as a subroutine.

- **Two DP Multi-RAG Algorithms for Varying Test Query Dependencies.** We propose two differentially private RAG algorithms for the multi-query setting, tailored to the degree of relevance among test-time queries. MURAG (Algorithm 1) uses a fixed relevance threshold across all queries and is sufficient to work well for settings where queries are independent and do not share relevant private documents. MURAG-ADA (Algorithm 2) allocates a small portion of the privacy budget to release a query-specific relevance threshold, enabling more efficient use of the budget when queries are related and share overlapping relevant documents.

- **Practical Multi-Query RAG with Non-Trivial Privacy Guarantees.** We evaluate our algorithms through extensive experiments on four question datasets (Natural Questions, Trivia Questions, ChatDoctor Questions, and MQuAKE Questions) and three LLMs (OPT-1.3B, Pythia-1.4B, and Mistral-7B). The external datasets include Wikipedia (commonly used in standard RAG setups) and ChatDoctor with QA pairs between patients and doctors, reflecting privacy-sensitive applications. Empirical results show that both methods can answer hundreds of queries under a towtal privacy budget of $\varepsilon \approx 10$ while achieving a reasonable utility, reducing privacy budget consumption by up to $100\times$ compared to the baseline DP RAG method that achieves the comparable utility. To the best of our knowledge, these are the first DP algorithms to achieve both non-trivial privacy guarantees and practical utility in the multi-query RAG setting.

## 2 Preliminaries

**Notation.** Let $\mathcal{V}$ denote a finite vocabulary set, and let $x \in \mathcal{V}^*$ represent a prompt of arbitrary length. The document set with arbitrary size is denoted by $D = \{z_1, z_2, \ldots\}$, where each document $z_i \in \mathcal{V}^*$. For convenience, we define the document space as $\mathcal{Z}$.

**Differential Privacy.** Given dataspace $\mathcal{X}$, two datasets $D, D' \in \mathcal{X}^*$ are said to be neighboring if they differ by at most one element. In this work, we focus on document-level privacy, where the data universe is given by $\mathcal{V}^*$.

**Definition 1** (Differential Privacy [Dwork et al., 2006b]). *A randomized algorithm $\mathcal{M} : \mathcal{X}^* \to \Omega$ is said to satisfy $(\varepsilon, \delta)$-differential privacy if for all neighboring datasets $X, X' \in \mathcal{X}^*$ and for all measurable subsets $O \subseteq \Omega$, we have*

$$\mathbb{P}(\mathcal{M}(X) \in O) \leq e^\varepsilon \mathbb{P}(\mathcal{M}(X') \in O) + \delta.$$

**Definition 2** (Rényi Differential Privacy [Mironov, 2017]). *A randomized algorithm $\mathcal{M} : \mathcal{X}^* \to \Omega$ is said to satisfy $(\alpha, \varepsilon)$-Rényi Differential Privacy (RDP) if for all neighboring datasets $X, X' \in \mathcal{X}^*$, the Rényi divergence of order $\alpha > 0$ between $\mathcal{M}(X)$ and $\mathcal{M}(X')$ is at most $\varepsilon$, i.e.,*

$$D_\alpha(\mathcal{M}(X)\|\mathcal{M}(X')) \leq \varepsilon.$$

We may also consider *individual-level* RDP, where the Rényi divergence is evaluated for neighboring datasets that differ on a particular data point $z_i$. We use $\mathcal{S}(z_i, n)$ to denote the set of neighboring dataset pairs $(S, \tilde{S})$ such that $|S|, |\tilde{S}| < n$, and $z_i \in S \triangle \tilde{S}$—i.e., exactly one of the datasets contains $z_i$.

**Definition 3** (Individual Rényi Differential Privacy). *A randomized algorithm $\mathcal{M} : \mathcal{X}^* \to \Omega$ satisfies $(\alpha, \varepsilon)$-individual RDP at point $z_i$ if for all datasets $X, X' \in \mathcal{S}(z_i, n)$, we have*

$$D_\alpha(\mathcal{M}(X)\|\mathcal{M}(X')) \leq \varepsilon.$$

A privacy filter is a stopping time that monitors the cumulative privacy loss and terminates the algorithm once the total privacy budget is exhausted, thereby ensuring that the prescribed privacy guarantees are not violated. We now introduce the definitions of (individual) privacy filters in the context of Rényi Differential Privacy.

**Definition 4** ((Individual) Rényi Differential Privacy Filters [Feldman and Zrnic, 2021]). *We say that a random variable $\mathcal{F}_{\alpha,B} : \Omega^* \to \{\mathrm{CONT}, \mathrm{HALT}\}$ is a privacy filter for $(\alpha, B)$-RDP if it halts the execution of an algorithm before its accumulated (individual) privacy loss exceeds $B$ (measured in $\alpha$-Rényi divergence).*

## 3 Problem Setting

We study retrieval-augmented generation (RAG) over a sensitive external dataset $D$. Given a user prompt $x \in \mathcal{V}^*$, a retrieval function $R$ selects the top-$k$ relevant documents $D_x = R(x, D; k)$ from $D$, and a decoder-only LLM then generates a response conditioned on both $x$ and $D_x$ using greedy decoding. The dataset $D$ contains individual records, each corresponding to a single person's private information. We adopt a *realistic threat model* where the adversary cannot directly access $D$ but may issue arbitrary prompts $x$ and observe the system's outputs. The underlying LLM is assumed to be public; the privacy risk arises solely from the retrieval over $D$.

Our objective is to design a *differentially private RAG algorithm* that answers a sequence of *online queries* $\{q_1, \ldots, q_T\}$ while protecting the privacy of $D$. Specifically, given the private dataset $D$, a public LLM, and a total privacy budget $\varepsilon$, we seek an algorithm $\mathcal{A}(D, \{q_1, \ldots, q_T\}, \mathrm{LLM}, \varepsilon)$ that generates high-utility responses while guaranteeing $\varepsilon$-differential privacy with respect to the external document dataset $D$.

## 4 Methodology

### 4.1 Technical Overview

**Document-level Privacy Accounting through Individual Privacy Filters.** In retrieval-augmented generation, each query typically accesses only a small (relative to the whole external dataset $D$), query-specific subset of relevant documents. This "sparsity" means most documents are retrieved infrequently. We exploit this by applying an individual-level privacy filter that tracks the cumulative privacy loss per document and halts access once its budget is exhausted. Since a document incurs privacy loss only when retrieved, this approach ensures privacy accounting scales with retrieval frequency rather than the total number of queries.

**Screening Relevant Documents via Adaptive Thresholding.** Applying RAG directly over the entire dataset negates individual privacy filters, as all documents contribute to privacy loss. To mitigate this, MURAG uses a global threshold $\tau$: only documents with relevance scores above $\tau$ are retrieved and charged privacy cost. Once a document's budget is exhausted, it is excluded from future responses. However, a fixed $\tau$ can be suboptimal due to varying relevance distributions across queries, leading to inefficient budget use. For example, a poorly calibrated $\tau$ may retrieve low-value documents early or omit high-value ones later, leading to inefficient budget use and degraded utility over time. To address this, we propose MURAG-ADA, which privately releases a query-specific threshold $\tau_t$ tailored to each query's relevance distribution. By combining individual privacy accounting with private release of cumulative statistics, MURAG-ADAselectively retrieves high-relevance documents, reducing unnecessary budget consumption on irrelevant documents and potentially preserving utility across other queries.

After screening relevant documents, we apply a single-query DP-RAG algorithm to generate the response. As shown in Algorithms 1 and 2, our multi-query framework is modular and supports any private single-query RAG method – that is, an algorithm that ensures DP with respect to the retrieved document set. In this paper, we instantiate it with a pure DP variant of the single-query DP-RAG algorithm from Koga et al. [2024], detailed in Algorithm 5.

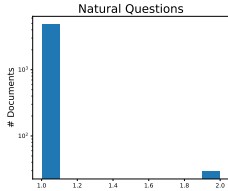 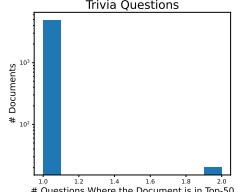 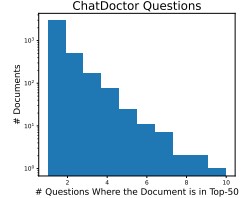 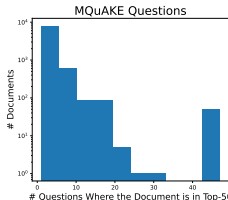

Figure 1: Histogram of how many questions each document appears in among the top-$K$ retrieved results ($K = 50$). The x-axis indicates the number of questions, and the y-axis shows the number of such documents. We show the histogram for four datasets.

## 4.2 DP RAG with fixed threshold

In MuRAG, we set a fixed threshold $\tau$ on the relevance score, which can be either public or privately estimated using a minimal portion of the privacy budget. This approach performs well when relevance score distributions are consistent across queries, as is the case for datasets such as Natural Questions and Trivia Questions. Implementation details are provided in Algorithm 1.

---

**Algorithm 1:** MuRAG: Differentially Private Multi-Query Retrieval-Augmented Generation

> **Input:** Private external dataset $D$, a sequence of online queries $\{q_1, q_2, \cdots, q_T\}$, per query privacy budget $\varepsilon_q$, per query retrieved number of documents $k$, maximum retrieval times per document $M$, relevance score threshold $\tau$
> **Set:** Individual privacy budget for each document $z \in D$: $\mathcal{E}(z) = M \cdot \varepsilon_q$,
> 1 **for** $t = 1, ..., T$ **do**
> 2     $A_t = \{z \in D \mid \varepsilon_i \geq \varepsilon_q\}$             ▷ Update active document set
> 3     $D_{q_t} = \{z \in A_t \mid r(z, q_t) > \tau\}$         ▷ Find relevant documents
> 4     **for** $z \in D_{q_t}$ **do**
> 5        $\mathcal{E}(z) = \mathcal{E}(z) - \varepsilon_q$          ▷ Update remaining privacy budget
> 6     $D_{q_t}^k = \text{Top-K}(D_{q_t}, k, r(\cdot, q_t))$
> 7     $a_t = \text{DP-RAG}(x, D_{q_t}^k, \text{LLM}, \varepsilon_q)$     ▷ Answer for question $q_t$ using Algo. 5
> 8 **return** $(a_1, a_2, \ldots, a_T)$

---

**Lemma 1** (Privacy guarantee of Algorithm 1). MuRAG *satisfies $\varepsilon$-differential privacy if, for every document $z \in D$, the ex-ante individual privacy budget is at most $\varepsilon$.*

## 4.3 DP RAG with Adaptive Threshold

To address the limitations of using a static threshold, we propose releasing a query-specific threshold corresponding to the top-$K$ relevance scores for each query. Specifically, we first discretize the similarity scores into bins and then iteratively release noisy prefix sums until the cumulative count exceeds $K$. Additional implementation details are provided in Algorithm 2. As we demonstrate in the experimental section, this approach yields improved utility on the ChatDoctor and MQuAKE datasets.

**Lemma 2** (Privacy guarantee of Algorithm 2). MuRAG-ADA *satisfies $\varepsilon$-differential privacy if, for every document $z \in D$, the ex-ante individual privacy budget is at most $\varepsilon$.*

# 5 Experiment

## 5.1 Dataset and Model set-up

**Question datasets in two categories.** We evaluate our methods on four question datasets: *Natural Questions*, *Trivia Questions*, *ChatDoctor Questions*, and *MQuAKE Questions*. Natural Questions [Kwiatkowski et al., 2019] and Trivia Questions [Joshi et al., 2017] are standard benchmarks for evaluating RAG systems and have been used in prior work on per-query DP for RAG [Koga et al., 2024]. Following their setup, we randomly subsample 100 questions from each dataset to

**Algorithm 2:** MURAG-ADA: DP Multi-Query RAG with adaptive threshold

---

**Input:** Private external dataset $D$, a sequence of online queries $\{q_1, q_2, \cdots, q_T\}$, per query budget $\varepsilon_q$, per query retrieved number of documents $k$, maximum retrieval times per document $M$, initial relevance threshold $\tau$

**Set:** Individual privacy budget for each document $z \in D$: $\mathcal{E}(z) = M \cdot \varepsilon_q$, privacy budget allocation: $\varepsilon_q = \varepsilon_{\mathrm{thr}} + \varepsilon_{\mathrm{RAG}}$

**Require:** Discretization on similarity scores $[a_i, b_i]_{i=1}^B$

**1** **for** $t = 1, ..., T$ **do**

    /* Release prefix sum */

**2**     $\tilde{s} = 0, A_t = \phi$

**3**     **for** $i = 1, \ldots, B$ **do**

**4**         $A_t^{(i)} = \{z \in D \mid r(z, q_t) \in [a_i, b_i] \text{ and } \mathcal{E}(z) \geq \varepsilon_{\mathrm{thr}}\}$

**5**         $\tilde{s} = \tilde{s} + |A_t^{(i)}| + \mathrm{Lap}(1/\varepsilon_{\mathrm{thr}})$

**6**         $A_t = A_t \cup A_t^{(t)}$

**7**         **for** $z \in A_t^{(i)}$ **do**

**8**             $\mathcal{E}(z) = \mathcal{E}(z) - \varepsilon_{\mathrm{thr}}$

**9**         **if** $\tilde{s} \geq K$ **then**

**10**            Halt

    /* RAG */

**11**     $A_t' = \{z \in A_t \mid \mathcal{E}(z) \geq \varepsilon_{\mathrm{RAG}}\}$

**12**     $D_{q_t}^k = \mathrm{TOP\text{-}K}(A_t', k, r(\cdot, q_t))$

**13**     $a_t = \mathrm{DP\text{-}RAG}(x, D_{q_t}^k, \mathrm{LLM}, \tau_t, \varepsilon_{\mathrm{RAG}})$         ▷ `Answer` $q_t$ `using Algo. 5`

**14**     **for** $z \in A_t'$ **do**

**15**         $\mathcal{E}(z) = \mathcal{E}(z) - \varepsilon_{\mathrm{RAG}}$         ▷ `Update remaining budget`

**16** **return** $(a_1, a_2, \ldots, a_T)$

---

reduce computational overhead. Chatdoctor Questions [Li et al., 2023] consist of QA interactions between patients and doctors in the healthcare domain. We sample 100 patient questions from the original dataset as our test set. MQuAKE Questions [Zhong et al.] contain sequences of semantically related single-hop questions that collectively form multi-hop reasoning chains. We select 100 such sequences, resulting in a test set of 400 individual questions.

To better analyze the performance differences between MURAG and MURAG-ADA, we categorize the four datasets into two types: *independent question sets* and *dependent question sets*. As discussed in Section 4.3, we expect MURAG-ADA to be particularly effective in datasets where questions are semantically related and share overlapping relevant documents, while offering limited benefit in datasets where questions are unrelated and retrieve disjoint sets of documents.

To support this categorization, we plot histograms showing how frequently each document appears in the top-$K$ retrieved results ($K = 50$) across questions. As shown in Figure 1, we observe that in *Natural Questions* and *Trivia Questions*, most documents are retrieved for only one or two questions. This indicates minimal overlap in relevant documents, and we therefore categorize them as *independent question sets*. In contrast, in *ChatDoctor Questions* and *MQuAKE Questions*, many documents are shared across multiple questions, suggesting substantial overlap in relevance. We categorize these as *dependent question sets*.

**External datasets reflecting both standard and privacy-sensitive settings.** For Natural Questions, Trivia Questions, and MQuAKE Questions, we use Wikipedia as the external knowledge source following the standard RAG setup [Chen et al., 2017, Lewis et al., 2020]. For ChatDoctor Questions, the external dataset consists of the remaining QA pairs from the original ChatDoctor dataset, excluding the 100 patient questions used for testing. This setup reflects a realistic privacy-sensitive application, where the external corpus contains inherently private information.

**QA evaluation metric.** For Natural Questions, Trivia Questions and MQuAKE Questions, the datasets provide a list of all acceptable correct answers for each question. Following the evaluation protocol of Koga et al. [2024], we use the *Match Accuracy* metric: a prediction is scored as 1 if it contains any correct answer, and 0 otherwise. For Chatdoctor Questions, we adopt the evaluation metric from the original dataset paper, using the F1 score of BERTScore [Zhang et al., 2020] to measure semantic similarity between the predicted response and the ground-truth answer.

**Model set-up.** Our RAG pipeline integrates three pre-trained LLMs: OPT-1.3B [Zhang et al., 2022], Pythia-1.4B [Biderman et al., 2023], and Mistral-7B [Jiang et al., 2023]. For document retrieval, we use the Dense Passage Retriever (DPR) [Karpukhin et al., 2020] to compute dense query-document relevance scores.

## 5.2 Method Set-up

**Baseline methods.** We compare our two proposed methods with three baselines. The first is DP-MULTI-RAG (Algorithm 4), which applies the per-question DP RAG method, DPSparseVoteRAG, independently to each query and uses the standard sequential composition theorem [Dwork et al., 2006a] to compute the overall privacy guarantee. The other two are non-private baselines: Non-RAG, which generates answers using the pretrained LLM without retrieval, and Non-Private-RAG, which performs retrieval-augmented generation without any privacy mechanism.

**Privacy budget setup for DP algorithms.** Following the setup in Koga et al. [2024], we vary the per-query RAG privacy budget $\varepsilon_q \in 2, 5, 10, 15, 20, 30, 40$ to explore the privacy-utility trade-off. For DP-MULTI-RAG, the total privacy budget is $T \cdot \varepsilon_q$, where $T$ is the number of questions. For MURAG and MURAG-ADA, the total budget is $M \cdot \varepsilon_q$, where $M$ is the number of retrieved documents with nonzero privacy loss. In our main results, we conservatively set $M = 1$ for a realistic privacy region. Moreover, $\varepsilon_{\text{thr}}$ is fixed as 1.0.

**Other hyperparameter settings.** All three DP algorithms rely on shared hyperparameters from DPSparseVoteRAG, including the number of retrieved documents $k$, the per-token privacy budget $\varepsilon_{\text{token}}$, and the SVT threshold $\tau_{\text{svt}}$. Following Koga et al. [2024], we evaluate each method under a grid of settings with $k \in 30, 40, 50$, $\varepsilon_{\text{token}} \in 0.5, 1.0, 2.0$, and $\tau_{\text{svt}} = k/2$. We report the best performance for each method over these configurations.

## 5.3 Results

Figure 2 presents the performance of our two proposed methods alongside three baselines across four datasets and three pretrained LLMs. To focus the analysis, we truncate the figure to show only the utility region above the Non-RAG baseline, i.e. where using a DP-protected RAG system offers a utility gain over simply using the pretrained LLM alone. This region is of primary interest, as it justifies the use of a private external dataset under differential privacy. Within this utility range, we compare different DP algorithms based on the amount of privacy budget they consume to achieve a given performance level.

**Comparing our methods with baselines.** Across all four datasets and three LLMs, both of our proposed methods consistently outperform the Non-RAG baseline at a total privacy budget of $\varepsilon = 10$. In contrast, the baseline method NAIVE-MULTI-RAG requires a significantly larger budget, exceeding $\varepsilon = 10^3$, to achieve comparable utility. These results demonstrate that our methods make differential privacy practical in the multi-query RAG setting, enabling strong performance while staying within a realistic privacy budget.

**Comparison between our two methods.** We observe a clear performance pattern between our two methods based on the dataset type. On the two *independent question sets* (*Natural Questions* and *Trivia Questions*) MURAG consistently outperforms MURAG-ADA. In contrast, on the *dependent question sets* (*ChatDoctor Questions* and *MQuAKE Questions*) MURAG-ADA shows superior performance. The improvement is especially pronounced on *MQuAKE Questions*, where MURAG performs only marginally better than the Non-RAG baseline, while MURAG-ADA yields a significant utility gain.

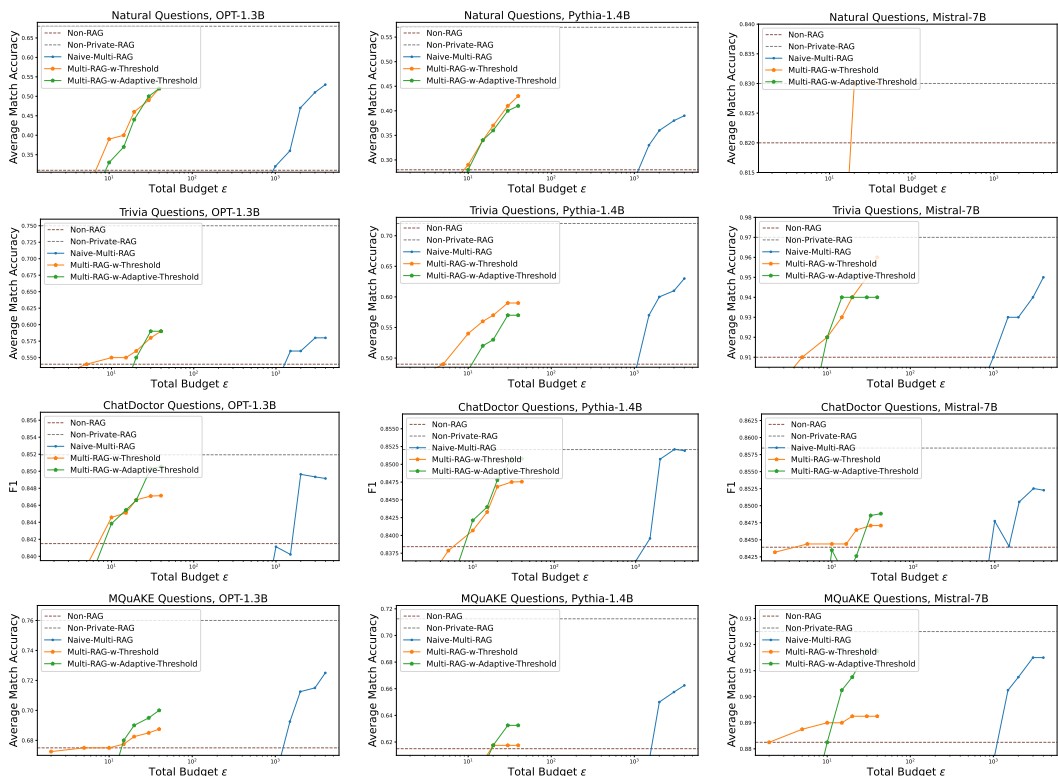

Figure 2: Utility-privacy tradeoffs of our two proposed methods (MuRAG and MuRAG-Ada) compared to three baselines across four question datasets and three pretrained LLMs. The plots are truncated to show only the region where the utility exceeds the Non-RAG baseline (i.e., the pretrained LLM without access to the external dataset), which is the regime of practical interest.

Table 1: Precision of retrieved documents under MuRAG and MuRAG-Ada, measured as the percentage of truly top-50 relevant documents among the retrieved. Results are reported for each dataset. MuRAG-Ada achieves significantly higher precision on the dependent question sets (ChatDoctor and MQuAKE), demonstrating its more efficient use of the privacy budget to save documents for later questions.

| | Independent Question Set | | Dependent Question Set | |
| --- | --- | --- | --- | --- |
| | Natural Questions | Trivia Questions | Chatdoctor Questions | MQuAKE Questions |
| MuRAG | 78.8% | 72.2% | 13.3% | 17.6% |
| MuRAG-Ada | 92.6% | 94.6% | 67.2% | 40.7% |
| MuRAG-Ada (non-private top-K-release) | 99.4% | 99.6% | 71.2% | 43.5% |

This discrepancy arises from the limitations of using a fixed retrieval threshold $\tau$ in MuRAG. Since different queries induce different distributions over relevance scores, a single global threshold can lead to imbalanced behavior: a threshold that retrieves sufficient relevant documents for one query may result in many low-relevance documents for another. These low-relevance documents still consume privacy budget without meaningfully improving the response, and may become unavailable for future queries where they are actually useful. This inefficiency is especially problematic in dependent question sets, where documents are frequently shared across queries.

To quantify this effect Table 1 reports the precision under both MuRAG and MuRAG-Ada, where the precision is defined as the percentage of truly top-50 documents among the retrieved documents for each question. We observe that precision under MuRAG is particularly low for *ChatDoctor Questions* and *MQuAKE Questions*, whereas MuRAG-Ada significantly improves retrieval precision on these datasets through its adaptive thresholds. This improvement in retrieval quality directly contributes to the superior performance of MuRAG-Ada in the setting of *dependent question set*.

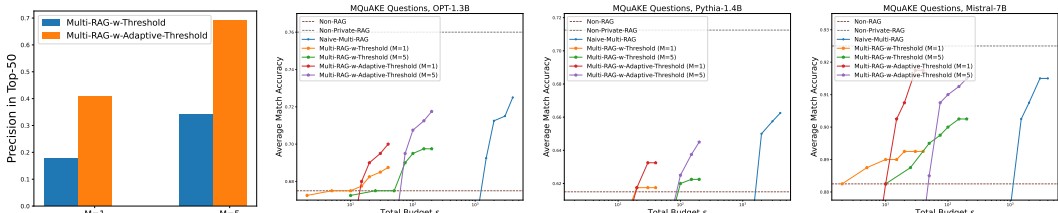

Figure 3: Comparison of $M = 1$ and $M = 5$ in the individual privacy accounting framework. Left plot shows the retrieval precisions of two methods with $M = 1, 5$. The right three plots show the trade-off between the QA performance and the $\varepsilon_{\text{total}}$ in DP. Increasing $M$ improves retrieval precision and RAG utility, but incurs a higher total privacy cost. $M = 1$ remains preferable for a stronger privacy-utility trade-off.

**Effect of different $M$ in the individual privacy accounting framework.** Both of our proposed methods include a hyperparameter $M$, which controls the maximum number of queries for which an individual document's privacy budget can be consumed. In our main results (Figure 2), we set $M = 1$ to ensure strict per-document privacy usage. However, this setting may limit utility: once a document is used for one query, it becomes unavailable for future queries, even if it would have been highly relevant. This limitation is evident in Table 1, where both methods exhibit low retrieval precision on the *MQuAKE Questions* dataset and this suggests that relevant documents were prematurely deactivated.

To better understand the impact of $M$, we evaluate our two methods with a larger value of $M = 5$. Figure 3 compares performance with $M = 1$ and $M = 5$. The left plot shows a substantial increase in Top-50 retrieval precision when using $M = 5$, indicating better access to relevant documents. This improvement translates into higher end-to-end RAG utility, as shown in the three plots on the right. However, increasing $M$ also leads to a higher total privacy cost ($\varepsilon_{\text{total}} = M \cdot \varepsilon_q$). Overall, while $M = 5$ enhances utility, we find that $M = 1$ still achieves the best privacy-utility trade-off under a practical privacy regime.

## 6 Discussion

**Why Privacy Filter rather than Amplification by Subsampling?** As surveyed in Section A.2, privacy amplification by subsampling [Balle et al., 2018, Wang et al., 2019, Zhu and Wang, 2019] is widely used in DP LLM applications, such as DP prompt tuning and DP in-context learning, to enhance generation quality. However, this technique is not well-suited for DP RAG:

- In prompt tuning, the goal is to learn a single task-specific prompt that can generalize to all future queries. In DP in-context learning, a small number of example inputs are selected under DP constraints and reused across queries. In contrast, our RAG setting does not allow for such "unified" prompts or examples: each test-time query requires retrieving and using query-specific documents, which must be handled privately, which makes individual privacy filter a more suitable choice.

- Moreover, in prompt tuning and in-context learning, all data points in the private dataset can meaningfully contribute to the learned prompt or selected example set. This property enables the use of subsampling-based amplification techniques in algorithm design. In RAG, however, only a sparse subset of documents in the large external corpus are relevant to any given query—most documents provide no utility.

These two key differences, the lack of reusable prompts and the sparsity of useful data, motivate the development of our new DP RAG algorithms using Rènyi filter rather than amplification by sampling.

**Leveraging Historical QA.** As shown in Table 1 and Figure 1, when the relevant documents for different questions exhibit significant overlap, the quality of answers to later questions degrades. This occurs because the documents required to answer the queries may exhaust their privacy budgets and are subsequently filtered out from the active set passed to the RAG algorithm. In the extreme case where a user repeatedly submits the same query, only the first response may retain high quality, while subsequent answers degrade due to the unavailability of relevant documents.

A potential remedy is to reuse historical answers as auxiliary documents in future queries. This can be done without incurring any additional privacy cost, owing to the post-processing property of differential privacy.

# 7 Conclusion

We proposed the first differentially private (DP) framework for retrieval-augmented generation (RAG) that supports answering multiple queries while protecting a sensitive external dataset. Our methods build on individual privacy accounting to overcome the limitations of the prior single-query DP-RAG system with the basic sequential composition. We introduced two algorithms—MURAG and MURAG-ADAdiffer in how they select documents for each query under DP guarantees. MURAG uses a fixed threshold for document retrieval, while MURAG-ADA adaptively adjusts the threshold per query to improve the efficiency of individual privacy accounting. Through comprehensive experiments on four question datasets and three LLMs, we demonstrated that both methods significantly reduce the privacy cost needed to outperform a Non-RAG baseline, achieving strong utility for answering $100$ questions under a realistic budget of $\varepsilon = 10$. We also showed that MURAG-ADA performs particularly well on datasets with overlapping document relevance across queries. This work highlights the importance of tailoring differential privacy mechanisms to the characteristics of the RAG setting. We hope our contributions provide a foundation for more practical and principled privacy-preserving RAG systems.

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

# A  Related Work

## A.1  Privacy-Preserving Retrieval Augmented Generation

Recent studies have revealed two main categories of privacy risks in retrieval-augmented generation (RAG) systems. The first is membership inference attacks (MIA)[Shokri et al., 2017], which aim to determine whether a specific document was included in the private external dataset. Research on MIA in RAG has explored adversarial prompt construction[Naseh et al., 2025, Liu et al., 2025, Anderson et al., 2024] and scoring mechanisms [Li et al., 2025] to increase inference success. The second type is data reconstruction attacks, which aim to recover the raw content of documents in the private dataset. These attacks have been explored via adversarial prompt design [Zhang et al., 2025, Zeng et al., 2024a, Jiang et al., 2024] as well as through data poisoning techniques that embed triggers to facilitate reconstruction [Peng et al., 2024]. Together, these works highlight the growing need for principled privacy-preserving algorithms for RAG.

Several approaches have been proposed to address these privacy risks. Most notably, Koga et al. [2024] introduced a differentially private (DP) RAG system that provides privacy guarantees for a single query. Other works [Yao and Li, Grislain, 2025] have studied how to release document identifiers under DP guarantees. However, none of these methods address the setting where multiple queries are issued, which is more representative of real-world usage. In addition to DP-based methods, there are empirical approaches that aim to enhance privacy without formal guarantees. Yao and Li propose a synthetic document generation technique, where retrieved documents are paraphrased before being passed to the generator. Zeng et al. [2024b] introduces a dataset privatization strategy that removes sensitive content across multiple documents prior to retrieval. While these methods offer promising empirical results, they lack worst-case guarantees and remain vulnerable to strong adversarial attacks. Finally, Cheng et al. [2024] explores a complementary privacy concern—protecting the user's query when interacting with a cloud-hosted RAG system. While important, this line of work addresses a different threat model than ours.

## A.2  Differential Privacy in Large Language Models

Beyond our focus on DP in RAG, DP has also been extensively studied in other LLM settings, including fine-tuning [Charles et al., 2024, Yu et al., 2021, Li et al., 2021], prompt tuning [Duan et al., 2023, Hong et al., 2024], and in-context learning [Tang et al., 2024, Wu et al., 2024]. Due to the differing nature of these tasks, the effective DP mechanisms vary significantly across settings. In pre-training and fine-tuning, the core challenge lies in optimizing the model parameters while maintaining stability against the noise introduced by DP algorithms. These settings are fundamentally different from RAG, where the goal is not to train the model but to ensure privacy during inference-time retrieval and generation. More closely related to our work are DP approaches for prompt tuning and in-context learning. However, the structural differences between these tasks and RAG result in distinct design considerations for differential privacy algorithms; please see Section 6 for more details.

## A.3  Individual Privacy Accounting and Privacy Filters

Individual privacy accounting focuses on tracking the privacy loss incurred by a single data point, often yielding tighter privacy bounds than traditional worst-case analyses over all neighboring datasets [Dwork et al., 2006b]. This line of work was initiated by Feldman and Zrnic [2021] in the context of Rényi Differential Privacy and was subsequently extended to Gaussian Differential Privacy [Dong et al., 2022] by Koskela et al. [2022]. For a comprehensive overview, we refer the reader to Feldman and Zrnic [2021, Section 1.2].

Building upon this framework, the concept of a privacy filter has been introduced as a general mechanism for adaptively enforcing privacy constraints. A privacy filter refers to a stopping rule that halts the execution of differentially private (DP) algorithms before the cumulative privacy loss exceeds a specified budget. Individual privacy filters, introduced by Feldman and Zrnic [2021] and further developed by Koskela et al. [2022], represent a specialized instantiation of this framework. These filters operate at the granularity of individual data points, terminating their participation once their respective privacy budgets are exhausted. For further details, we refer the reader to Rogers et al.

483 [2016], Feldman and Zrnic [2021], Koskela et al. [2022], Smith and Thakurta [2022], Whitehouse
484 et al. [2023].

## B Algorithms

486 We present additional algorithms that were omitted from the main paper for brevity.

### B.1 Top-k document selection

488 Algorithm 3 returns the top-$K$ documents from a dataset $D$ ranked by a score function $r$, padding
with empty strings if $|D| < K$ to ensure the output always has exactly $K$ elements.

---

**Algorithm 3:** TOP-K$(D, K, r)$

---

    **Input:** dataset $D$, sample size $K$, score function $r$

1 **if** $|D| \geq K$ **then**

2     $D^k \leftarrow$ top-$K$ documents from $D$ according to score function $r$     ▷ assume no ties

3 **else**

4     $D^k \leftarrow D \cup \{\text{""}\}^{k-|D|}$     ▷ Add empty string "" until document set has
    size $k$

5 **return** $D^k$

---

489

### B.2 Naive algorithm for DP Multi-Question RAG

491 Algorithm 4 serves as a baseline for handling multi-query DP RAG by composing a single-query
DP-RAG mechanism $T$ times.

---

**Algorithm 4:** NAIVE-MULTI-RAG

---

    **Input:** Private external dataset $D$, a sequence of queries $\{q_1, q_2, \cdots, q_T\}$, total privacy
        budget $\varepsilon$, per query budget $\varepsilon_q$

    **Require:** $\varepsilon \geq T \cdot \varepsilon_q$

1 **for** $t = 1, ..., T$ **do**

2     $a_t = \text{DP-RAG}(x, D, \text{LLM}, \varepsilon_q)$     ▷ Algo. 5

3 **return** $(a_1, a_2, \ldots, a_T)$

---

492

### B.3 DP-RAG for single question answering

494 Algorithm 5 is a variant of Koga et al. [2024, Algorithm 2], in which we replace the LimitedDomain
495 mechanism [Durfee and Rogers, 2019] with the exponential mechanism in the private token generation
496 step. This modification yields a more stringent pure-DP guarantee and results in a cleaner privacy
497 composition accounting result.

**Algorithm 5:** DP-RAG$(x, D, \text{LLM}, \varepsilon)$

---

    **Input:** Prompt $x$; external data source $D$; next-token generator
            $\text{LLM}(\text{prompt}, \text{doc} \mid \text{history})$; total budget $\varepsilon$;
    **Set:** Per-token privacy budget $\varepsilon_0$
    **Require:** maximum length of output tokens $T_{\max}$; number of voters $m$; retrievals per voter $k$;
            document retriever $R(\text{prompt}, \text{doc set}, \#\text{retrieved docs})$; threshold for voting $\theta$

**1**   $\varepsilon_{\text{Expo}} \leftarrow \varepsilon_0/2, \varepsilon_{\text{Lap}} \leftarrow \varepsilon_0/2$   ▷ split privacy budget for per token generation
**2**   $c \leftarrow \lfloor \varepsilon/\varepsilon_{\text{RAG}} \rfloor, \hat{\theta} \leftarrow \theta + \text{Lap}(2/\varepsilon_{\text{Lap}})$
**3**   $D_x \leftarrow R(x, D; mk)$                              ▷ retrieve $mk$ documents
**4**   $\mathcal{D}_x \leftarrow \{D_x^1, \ldots, D_x^m\}$        ▷ Partition $D_x$ into $m$ subsets uniformly random
**5**   **for** $t \leftarrow 1$ **to** $T_{\max}$ **do**
**6**       $y_t^{\text{non-RAG}} \leftarrow \text{LLM}(x, \text{""} \mid y_{<t})$
**7**       **for** $i \leftarrow 1$ **to** $m$ **do**
**8**           $y_t^{(i)} \leftarrow \text{LLM}(x, D_x^i \mid y_{<t})$
**9**       $\text{Hist}_t \leftarrow \text{Hist}(y_t^{(1)}, \ldots, y_t^{(m)})$                      ▷ $\text{Hist}_t \in \mathbb{N}^{|\mathcal{V}|}$
**10**      $\text{Count}_t \leftarrow \text{Hist}_t[\text{index} = y_t^{\text{non-RAG}}]$
**11**      **if** $\text{Count}_t + \text{Lap}(4/\varepsilon_{\text{Lap}}) \leq \hat{\theta}$ **then**
**12**          $y_t \leftarrow \text{expoMech}(\text{Hist}_t; \varepsilon_{\text{Expo}})$
**13**          $c \leftarrow c - 1$
**14**      **else**
**15**          $y_t \leftarrow y_t^{\text{non-RAG}}$
**16**      **if** $y_t = \langle \text{EOS} \rangle$ ***or*** $c = 0$ **then**
**17**          **return** $(y_1, \ldots, y_t)$
**18** **return** $(y_1, \ldots, y_{T_{\max}})$

---

## C   Proofs for Privacy Guarantee

### C.1   Privacy Guarantee for Algorithm 1

**Lemma** (Restatement of Lemma 1). MURAG *satisfies $\varepsilon$-differential privacy if, for every $z \in D$, the ex-ante individual privacy budget is at most $\varepsilon$.*

*Proof.* Since $\mathcal{E}(z) \leq \varepsilon$ for every $z \in D$, the privacy guarantee follows directly from Feldman and Zrnic [2021, Corollary 3.3].          □

### C.2   Privacy Guarantee for Algorithm 2

**Lemma** (Restatement of Lemma 2). MURAG-ADA *satisfies $\varepsilon$-differential privacy if, for every $z \in D$, the ex-ante individual privacy budget is at most $\varepsilon$.*

*Proof.* We first bound the individual privacy for the $t$-th prefix-sum release algorithm, denoted by $\mathcal{A}_t$. Consider $S, \tilde{S} \in \mathcal{S}(z_i, n)$, and without loss of generality, we assume $z_i \in S$. *Conditioned on the trajectory $r^{(t-1)}$ from the previous $t - 1$ rounds*, for any possible output sequence $b^{(q)} := (b_1, b_2, \ldots, b_q)$ with $q \leq B$, the interesting regime is that there exists $j \in [q]$ such that $z_i$ contributes to $b_j$; otherwise, we have $\mathcal{A}_t(S \mid r^{(t-1)}) \overset{\text{d}}{=} \mathcal{A}_t(\tilde{S} \mid r^{(t-1)})$. In the former case, we can perform the decomposition using Bayes' rule:

$$\log\left(\frac{\mathbb{P}(\mathcal{A}_t(S) = b^{(q)})}{\mathbb{P}(\mathcal{A}_t(\tilde{S}) = b^{(q)})}\right) = \underbrace{\log\left(\frac{\mathbb{P}(\mathcal{A}_t(S)[j+1:q] = b^{(j+1:q)} \mid b^{(j)})}{\mathbb{P}(\mathcal{A}_t(\tilde{S})[j+1:q] = b^{(j+1:q)} \mid b^{(j)})}\right)}_{(a)} + \underbrace{\log\left(\frac{\mathbb{P}(\mathcal{A}_t(S)[j] = b_j \mid b^{(j-1)})}{\mathbb{P}(\mathcal{A}_t(\tilde{S})[j] = b_j \mid b^{(j-1)})}\right)}_{(b)}$$

$$+ \underbrace{\log\left(\frac{\mathbb{P}(\mathcal{A}_t(S) = b^{(j-1)})}{\mathbb{P}(\mathcal{A}_t(\tilde{S}) = b^{(j-1)})}\right)}_{(c)}$$

$$\leq \varepsilon_{\mathrm{thr}}$$

Notice that every two bins are disjoint; thus, the consumption of the privacy budget is independent between two different data points. Therefore, $(a), (c) = 0$ and $(b) \leq \varepsilon_{\mathrm{thr}}$ by the privacy guarantee for single-query release.

Now we consider the RAG step. The interesting regime is when $z_i \in A'_t$. Then, by the composition theorem, the privacy loss of DP-RAG $\circ$ TOP-K is upper bounded by $\varepsilon_{\mathrm{RAG}}$.

In addition, we note that $\mathcal{E}(z_i)$ is a valid stopping time, since it updates the privacy budget after each invocation of the algorithms, and $z_i$ is only involved when its privacy budget is sufficient.

Thus, by Feldman and Zrnic [2021, Corollary 3.3], the privacy guarantee is given by $\mathcal{E}(z)$, which is upper bounded by $\varepsilon$. $\qquad\square$

