# OpenReview forum: "Beyond Per-Question Privacy: Multi-Query Differential Privacy for RAG Systems"
_NeurIPS.cc/2025/Workshop/Reliable_ML — NeurIPS 2025 - Reliable ML Workshop_

### Official Review · Reviewer_tEE3 · 2025-09-19
**Beyond Per-Question Privacy: Multi-Query Differential Privacy for RAG Systems — strong conceptual contribution (individual privacy filters for multi-query DP-RAG), promising empirical results, but theoretical precision, evaluation coverage, and reproducibility are lacking.**

**Rating:** 5
**Confidence:** 4

**Review:**

Summary

The paper addresses the gap between per-query DP guarantees and real-world multi-query usage in Retrieval-Augmented Generation (RAG). Existing methods (e.g., DPSparseVoteRAG) provide privacy for a single query but compose linearly across $T$ queries, exhausting budgets ($\varepsilon \approx 1000$ for 100 queries). The authors propose:
	•	MURAG: uses individual-level Rényi privacy filters so that each document accrues privacy cost only when retrieved, decoupling budget growth from $T$.
	•	MURAG-ADA: adds an adaptive, DP-released threshold for document relevance, improving retrieval precision and budget efficiency when queries overlap in relevant documents.

Experiments on four datasets (Natural Questions, TriviaQA, ChatDoctor, MQuAKE) with three LLMs (OPT-1.3B, Pythia-1.4B, Mistral-7B) show that both methods achieve strong QA utility for 100 queries under $\varepsilon = 10$, a 100× improvement over naïve composition. MURAG works best when queries are independent; MURAG-ADA excels when queries share documents.

⸻

Strengths
	•	Novelty: First principled multi-query DP framework for RAG, introducing per-document Rényi filters.
	•	Conceptual clarity: Clean modularity — any single-query DP-RAG subroutine can be plugged in.
	•	Algorithms tailored to query overlap regimes (MURAG vs. MURAG-ADA).
	•	Empirical evidence: Across multiple datasets, shows large improvements in utility–privacy trade-offs.
	•	Relevance: Addresses a highly practical gap in privacy-preserving RAG deployments.

⸻

Weaknesses / Limitations

Theory & definitions
	•	The formal privacy analysis is thin: Lemma proofs mainly restate filter properties; rigorous accounting for adaptive threshold release is underdeveloped.
	•	Assumption that retrieval scoring distributions are stable enough for fixed thresholds in MURAG is not justified.
	•	The $\varepsilon$ values reported ($\varepsilon=10$) are treated as “realistic” but lack calibration: no $(\varepsilon,\delta)$ trade-off discussion, no composition under Gaussian DP.

Methodology
	•	Experiments subsample only 100 queries per dataset; scalability to hundreds/thousands of queries is claimed but not tested.
	•	Utility metrics are narrow: only Match Accuracy and BERTScore-F1; no robustness analysis (adversarial prompts, noisy retrieval).
	•	Baselines are weak: compare only to per-query DPSparseVoteRAG; omit DP top-$k$ methods, subsampling-based amplification, or synthetic-data defenses.
	•	Privacy–utility reporting is limited: no ROC/PR curves, no per-document budget exhaustion analysis.
	•	Adaptive thresholds rely on noisy prefix sums; stability and sensitivity to binning not studied.

Reproducibility
	•	No code release; algorithmic details (Laplace/Gaussian scale, seed control, implementation of DP-RAG subroutine) are under-specified.
	•	Many hyperparameters (e.g., $M$, $k$, $\varepsilon_{\text{thr}}$ allocation) are chosen ad hoc without ablation.

⸻

Suggestions for Authors
	1.	Provide formal composition proofs for MURAG-ADA (threshold release + retrieval + generation).
	2.	Report $(\varepsilon,\delta)$ values under multiple DP formalisms (RDP, GDP). Clarify whether $\varepsilon=10$ corresponds to meaningful protection under adversarial query adaptation.
	3.	Test scalability beyond 100 queries, ideally to thousands, to show practical viability.
	4.	Add stronger baselines: DP top-$k$, privacy amplification by subsampling, synthetic privatization (e.g., paraphrase-based), energy-based defenses.
	5.	Extend evaluation metrics: calibration error (ECE), FPR@95%TPR, robustness under adaptive/adversarial queries.
	6.	Perform ablation on $M$, $\varepsilon_{\text{thr}}$, k, bin size, and document budget exhaustion curves.
	7.	Release code and configs to allow reproducibility and benchmarking.

---

### Official Review · Reviewer_iw51 · 2025-09-19

**Rating:** 6
**Confidence:** 2

**Review:**

**Summary:** This paper extends differential privacy in RAG systems from single queries to multiple queries using individual privacy accounting. The authors propose MURAG and MURAG-ADA algorithms that track privacy loss per document rather than per query, exploiting the sparsity of document retrieval in RAG. Experiments show the methods handle 100 queries with $\epsilon = 10$ while baselines require $\epsilon = 1000$ for comparable utility.

### **Strengths**

* The extension from single-query to multi-query DP-RAG is essential for real-world deployment. The authors correctly identify that naive composition quickly makes privacy guarantees meaningless, making this a well-motivated problem.

* The use of individual Renyi filters in the RAG context is technically sound and well-adapted to the problem structure. The insight that privacy accounting should scale with document retrieval frequency rather than total queries sounds feasible.

* The evaluation spans multiple datasets (including privacy-sensitive ChatDoctor), multiple LLMs, and provides good analysis of when each method works best.

* The 100$\times$ reduction in required privacy budget compared to naive composition makes multi-query DP-RAG practically feasible.

### **Areas for Improvement**

* **Limited theoretical analysis**: While the privacy guarantees are established through existing results (Feldman and Zrnic, 2021), the paper lacks deeper theoretical analysis of the privacy-utility tradeoffs specific to the RAG setting. The connection between retrieval sparsity and privacy efficiency could be formalized more rigorously.

* **Restrictive assumptions about document relevance**: The effectiveness relies heavily on the assumption that only a sparse subset of documents are relevant per query. The paper doesn't adequately address edge cases where this assumption breaks down or analyze robustness to variations in sparsity patterns.

* **Limited baseline comparisons**: The paper primarily compares against naive composition. Comparisons with other potential approaches (e.g., subsampling-based methods, alternative privacy accounting techniques) are needed.

* **Threshold selection methodology**: While MURAG-ADA addresses adaptive thresholding, the paper doesn't provide principled guidance for threshold selection in MURAG or analyze sensitivity to threshold choices.

* **Individual privacy accounting**: The application is technically correct, but the paper could better explain why existing subsampling-based amplification techniques aren't suitable beyond the brief discussion in Section 6.

* **Algorithm design**: The dependency on the specific single-query algorithm (Algorithm 5) could be better analyzed.

* **Selection of $M$**: In the main experiments, $M = 1$ is conservative but may underestimate the true utility potential. The brief analysis with $M = 5$ suggests more exploration of this parameter could be valuable.

* **Privacy budget allocation**: The split between $\epsilon$-th and $\epsilon$RAG in MURAG-ADA appears somewhat ad-hoc. More principled optimization of this allocation would improve.

### **Minor Issues**

- Figure 1 could be clearer with better axis labeling and captions
- Some notation is inconsistent (e.g., switching between D and Z for document space)
- The connection between retrieval precision (Table 1) and overall utility could be made more explicit
- Algorithm 2 has some formatting inconsistencies in the pseudocode